# The salty tango of brine composition and UV photochemistry effects on *Halobacterium salinarum* cell envelope biosignature preservation

Lucas Bourmancé [1], Arul Marie[1], Rémy Puppo[1], Sébastien Brûlé[2], Philippe Schaeffer [3], Maud Toupet[1], Ruben Nitsche [4], Andreas Elsaesser [4] & Adrienne Kish [1] ✉

Hypersaline environments, including brines and brine inclusions of evaporite crystals, are currently of great interest due to their unique preservation properties for the search for terrestrial and potentially extraterrestrial biosignatures of ancient life. However, much is still unclear about the specific effects that dictate the preservation properties of brines. Here we present the first insights into the preservation of cell envelope fragments in brines, characterizing the relative contributions of brine composition, UV photochemistry, and cellular macromolecules on biosignature preservation. Cell envelopes from the model halophile *Halobacterium salinarum* were used to simulate dead microbial cellular remains in hypersaline environments based on life as we currently know it. Using different Early Earth and Mars analogue brines, we show that acidic and NaCl-dominated brine compositions are more predisposed to preserving complex biosignatures from UV degradation, but that the composition of the biological material also influences this preservation. Furthermore, a combinatory effect between chaotropicity and photochemistry occurs, with the relative importance of each being brine-specific. These results provide an experimental framework for biosignature detection in hypersaline environments, emphasizing the need for laboratory simulations to evaluate preservation properties of each potential brine environment, on Earth and elsewhere in the solar system.

Hypersaline environments can be found in many places on Earth[1]. In saturated brines, microbial diversity tends to be highly reduced and dominated by extremely halophilic archaea[2]. Increase in temperature and reduction of relative humidity in brines lead to the precipitation and the formation of salt crystals such as halite in NaCl-dominated systems. During the evaporation process, fluid inclusions[3] derived from the original brine are trapped inside the crystalline matrix, along with any microorganisms present. NaCl is one of the last salts to precipitate in such systems, thus retaining the final traces of brine before complete evaporation[4]. Evidence of halophilic archaea from the genus *Halobacterium* preserved in halite fluid inclusions has been well documented[5]. As those crystals have been preliminarily dated through their geological context to be up to millions of years old[6,7], preserved biomolecules within

the fluid inclusions may serve as potential biosignatures of past ancient life on Earth.

Over such prolonged periods within brine inclusions, cell lysis will eventually occur producing cell envelope fragments. These fragments, composed of membrane lipids, embedded cell envelope proteins, and for *Halobacterium*, cell wall surface layer proteins, constitute biosignatures of interest. Membrane lipids have been described as high priority candidates for biosignatures because of the long-term stability properties of hydrocarbons[8–12]. Moreover, carotenoid lipid pigments (bacterioruberin) found in haloarchaeal diether lipid bilayer membranes possess passive antioxidant properties, thus offering protection to surrounding molecules from degrading oxidation reaction[13]. Cell envelope proteins are considered as lower interest biosignature targets as they are usually less stable.

¹Unité Molécules de Communication et Adaptation des Microorganismes (MCAM), Muséum National d'Histoire Naturelle (MNHN), CNRS, Paris, France. ²Institut Pasteur, Université Paris Cité, Plateforme de Biophysique Moléculaire, Paris, France. ³Institut de Chimie de Strasbourg, Université de Strasbourg, CNRS, Strasbourg, France. ⁴Experimental Biophysics and Space Sciences, Institute of Experimental Physics, Freie Universität Berlin, Berlin, Germany. ✉e-mail: adrienne.kish@mnhn.fr

However, hypersaline conditions may increase their preservation. In brines, water activity is highly reduced[14] whereas density and viscosity tend to increase, affecting both chemical reactions and degradation rates[15–17]. Furthermore, halophilic protein adaptations to molar intracellular salt concentrations as part of the 'salt-in' homeosmotic strategy may augment their preservation potential. These include the presence of an increased number of acidic amino acid residues on protein surfaces, smaller hydrophobic patches, and salt bridges[18–21], which have been demonstrated to provide structural protection from chaotropic agents, temperature and solvents[22]. Therefore, unlike their non-halophilic counterparts, proteins from halophiles are important candidates for biosignatures. Finally, the outermost archaeal cell structure in contact with the exterior environment, the proteinaceous surface layer (S-layer), in particular, has shown interesting mineral-binding preservation properties. Previous studies on thermophilic and methanogenic archaea (*Sulfolobus acidocaldarius*, *Methanocaldococcus jannaschii*, *Pyrococcus abyssi*[23–26]) have shown S-layer mineralisation after interactions with environmental (metal) ions, supporting it as a good biosignature candidate depending on ionic interactions. The stability of S-layer interactions within the *Halobacterium* cell envelope, both between protein monomers and associations with lipids, is also dependent on ionic interactions[27]. Both low salinity (NaCl) and chaotropic agents such as $Li^+$ and $Mg^{2+}$ can disrupt S-layer stability. In contrast, species adapted to more chaotropic magnesium-containing brines such as *Haloferax volcanii* require roughly 100 mM $Mg^{2+}$ to stabilize their S-layer structure[28]. It is important to note that S-layer stability differs between whole cells and cell envelope extracts[20], likely due to ion accessibility.

The ionic composition of brines is significant for protein preservation by exerting a direct influence on the chemical stability of macromolecular interactions. The dominant salts in a given brine can be classified as either chaotropic or kosmotropic, indicating their capacity to respectively destabilize macromolecules by promoting indiscriminate, large-scale instability or stabilize macromolecules by rigidifying cellular structures[29–32]. Using a standard agar gelation point assay[33], sodium-based brines are classified as slightly (NaCl) to moderately (NaSO$_4$) kosmotropic, whereas magnesium salts can be either kosmotropic (MgSO$_4$) or chaotropic (MgCl$_2$) depending on the anion. Previous studies have assessed the chaotropic effects of various salts on the structure of isolated non-halophilic and halophilic proteins[34–36]. Those studies have shown that Na, Mg or Ca ions are able to change α-helix and β-sheet content and tryptophan side chain exposure. However, much is still unclear about the implications of this effect both for mixed macromolecular complexes such as cell envelopes and for complex saturated brine solutions.

Current bioanalytical techniques are rarely compatible with saturated salt conditions, requiring extensive optimisation to render them suitable for analyses of evaporites. Advancements have recently been made[37,38] to analyse entrapped haloarchaea with proteomics and transcriptomics approaches along with improved crystal sterilisation procedures. Additionally, previous studies have focused on chaotropicity effects on halophilic and non-halophilic proteins; however, these investigations were typically conducted at lower salt concentrations and primarily involved soluble proteins[35,36,39]. Furthermore, for samples with multiple brine compositions with varying physiochemical properties (pH, water activity, viscosity), it is essential to develop compatible analytical methods for reliable comparison. This will help to determine methods that are compatible with specific environments, but also biomolecules that are most promising biosignatures and biomarkers[40]. Constraining the conditions of biomolecule preservation in fluid inclusions and developing new methodology for more accurate analysis is also of great exobiological interest.

High salt environments have been discovered in many other planetary bodies in the solar system including Mars, and the icy moons of Europa and Enceladus as subsurface salty oceans[41]. While halite crystals have been identified at the surface of Mars[42,43], early Mars was mostly dominated by sulphate hypersaline lakes[44,45]. However, a diversity of hypersaline environments existed on the planet covering drastically different ionic compositions[46,47]. Moreover, hypersaline environments on Earth have known modifications throughout geological times. Therefore, studying the preservation of biomolecules of interest in media having different ionic compositions is essential to constraining preservative environments. Furthermore, photochemistry plays an important role for brine inclusions located at the surface of planets or moons under exposure to full spectrum UV and other types of radiation. According to a specific ionic composition, unique photochemistry will occur and lead to specific degradation of organic matter caused by reactive species and radicals[48].

The current work presented in this paper aims to investigate the relative contributions of brine chaotropicity and UV photochemistry to the preservation of cell envelope as biosignatures in hypersaline environments and simulated salt brine inclusions. The results obtained are of relevance to the detection of any ancient biosignatures preserved in brines on Earth and potential biosignatures on Mars.

## Results

To determine the effect of different brine inclusion compositions on the preservation of cell envelope proteins, their effect on proteins structural stability were first investigated.

### Assessment of the chao-kosmotropic effect of mixed brine on cell envelope structural stability

Calculated mixed brine chaotropicity depends on composition. Chaotropicity differences of the mixed brine solutions were evaluated using the agar gelation point assay previously established for use with single-salt solutions[33]. A microplate-based assay (Supplementary Fig. 1, Table 1) was performed. Relative to the control salt BSS, the brine M1 demonstrated kosmotropic activity, while the remaining brines exhibited chaotropic behaviour, with the exception of the E1 brine, which showed no effect. The M3 and M4 brines showed pronounced chaotropicity, which could not be measured by the microplate assay due to temperature limitations. Instead, a visual "by-eye" evaluation method was employed. The effects of certain brines (M1, M2, and M3) used in this study have been previously assessed in earlier work[14]. Although both M1 and M2 once again exhibited kosmotropic activity relatively to the 1.5% agar solution (H$_2$O), they did not display identical levels of activity. Furthermore, M3 was also identified as a kosmotrope, rather than a chaotrope. It is hypothesized that these observed discrepancies may be attributed to variations in the agar used or the precision of the plate reader employed in the measurements. Finally, this method exclusively demonstrates the implications of chao-kosmotropic effects on a single molecule. However, it is important to underscore that such effects may differ when applied to more complex structures, such as cell envelope fragments.

Brine chaotropicity validated by effects on cell envelope protein structural stability. NanoDSF was employed to assess the direct impact of various brines on the structural integrity of *Halobacterium* cell envelope fragments by monitoring the intrinsic fluorescence of embedded protein Trp residues (Fig. 1). Cell envelope extracts incubated with the control salt BSS displayed a clear peak, relative to the transition from the folded to unfolded state, and a melting temperature (Tm) of 81.5 °C was observed (Fig. 1a). Similarly, clear transitions were detected for the BSS-LS and E1 brines (Fig. 1b, f). In contrast, brines E2 and M2 (Fig. 1d, g) exhibited less pronounced transitions from the folded to unfolded states, with estimated Tm values around 50 °C. The M4 brine (Fig. 1e) did not show any observable transition. It is hypothesized that the chaotropic nature of brines E2, M2, and M4, as assessed by the agar gel-point assay, induced partial or complete denaturation of cell envelope proteins during the incubation process prior to measurement. This likely resulted in minimal or even a lack of transitions between folded and unfolded states during the DSF assay. This phenomenon was anticipated for the M4 brine, given the well-characterized strong chaotropic properties of perchlorate. Upon incubation of cell envelope extracts in this brine, a rapid

**Table 1 | Results from chao-kosmotropicity agar gel-point assays**

| Brines | Gelation point (°C) | Compared to H₂O | | Compared to BSS | |
|---|---|---|---|---|---|
| | | Extrapolated brine gel point increase/decrease (°C) | Chaotropic(+) or Kosmotropic (−) Activity (kJ.kg⁻¹) | Extrapolated brine gel point increase/decrease (°C) | Chaotropic(+) or Kosmotropic (−) Activity (kJ.kg⁻¹) |
| Agar 1.5% (H₂O) | 37 | N/A | N/A | −6.5 | 26.98 |
| M1 | 44 | 21 | −87.15 | 10.5 | −43.58 |
| E1 | 40.5 | 10.5 | −43.58 | 0 | 0 |
| BSS | 40.5 | 10.5 | −43.58 | N/A | N/A |
| BSS-LS | 40 | 9 | −37.35 | −1.5 | 6.23 |
| E2 | 37 | 0 | 0 | −10.5 | 43.58 |
| M2 | 37.5 | −1.5 | 6.23 | −9 | 37.35 |
| M3* | 4 | −99 | 410.85 | −109.5 | 454.43 |
| M4* | < 4 | < −99 | < 410.85 | < −109.5 | < 454.43 |

Chaotropicity values are calculated knowing that the heat capacity of a 1.5% (w/v) agar gel is 4.15 kJ.kg⁻¹. °C⁻¹. Negative values indicate chaotropic activity, positive values indicate kosmotropic activity. (*) indicates incompatibility with the microplate assay, necessitating the assessment of the gelation point through a visual "by-eye" observation method. *N/A* Not Applicable.

decrease in the intensity of their red colour was observed, indicating the destabilisation or degradation of carotenoid lipid pigments responsible for the characteristic red-orange colouration of haloarchaea. To evaluate this chaotropic effect more precisely over time, cell envelope extracts were resuspended in BSS, M2, and M4 brines and immediately analysed. However, the effect of these solutions on membrane protein structures occurs too rapidly to be monitored in real time by NanoDSF (see Supplementary Fig. 2).

The M1 brine displayed a first transition around 60–70 °C, but also a more distinct transition beginning at approximately 90 °C, suggesting a kosmotropic effect on cell envelope proteins. A larger temperature gradient would be necessary to accurately assess the kosmotropic nature of the M1 brine, but could not be achieved due to temperature restriction of the nano-DSF machine used.

Overall, the NanoDSF results corroborate the classification of the brines based on their chaotropic and kosmotropic effects, as determined by the agar gelation point assay, except for E1, which displays a slight kosmotropic activity in this analysis. Thus, in this case, the influence of the brines on simple agar molecules parallels their effect on complex cell envelope proteins within the cell envelope structure, offering a proxy measurement of biosignature degradation.

Tryptophan residues fluorescence can also be exploited more directly as a marker for structural changes. By measuring the emission spectra following excitation of Trp residues within a complex cell envelope sample at 295 nm, downward shifts (blue shift) in emission wavelength indicate a relocation of Trp residues towards protein interiors, whereas shifts toward higher wavelengths (red shift) indicate increased exposure of Trp residues to the solvent, in this case the brine environment. This technique thus provides more precise information about the salt effects on Trp structure.

Cell envelopes incubated in each of the early Earth and early Mars brine solutions were compared to the control (BSS). Three distinct groups were categorized: (1) brines E1, M1, BSS-LS, and E2 did not induce significant structural changes on the tryptophan residues, (2) brine M4 caused significant exposure of tryptophan residues to the brine environment, and (3) the M2 brine resulted in more buried tryptophan residues (Table 2). These findings bring nuance regarding the chaotropic nature of brines E2, M2 and M4 and underline the complexity of defining chaotropicity, with three chaotropic brines exhibiting three different effects.

**Consequence of brine chaotropicity on cell envelope protein biosignature identification.** The LC-MS/MS identification profiles of cell envelope proteins of cell envelope extracts incubated in the different brines were compared to those incubated in the control salt BSS (Fig. 2, Supplementary Data 1).

Analysis revealed that the percentage of cell envelope proteins identified was impacted following incubation in all hypersaline conditions compared to BSS. Notably, the M2 and M4 brines induced more substantial changes, with respectively 19% and 25% of cell envelope proteins being affected by the incubation (Fig. 2a). Moreover, cytosolic contaminants were more impacted by the ionic composition of the brines than the cell envelope proteins (Fig. 2b).

The presented results therefore suggest a strong effect of ionic composition on the cell envelope structural stability with brines exhibiting great chao-kosmotropic activities, especially with regards to the cell envelope proteins.

**Brine composition affects protein preservation following UV irradiation**

To investigate preservation properties of brines and their respective chao-kosmotropic activities in a photooxidative environment such as on Mars, irradiation experiments were conducted with cell envelope extracts being incubated in the different solutions.

This effect was assessed by evaluating cell envelope proteins degradation upon ultraviolet (UV) irradiation using nano-LC/MS-MS analysis (Fig. 3, Supplementary Data 2). Two groups of brines were identified based on the percentage of cell envelope proteins degraded after exposure (Fig. 3a–i). Proteins were better preserved in brines BSS, BSS-LS and M2, with only ~5% of cell envelope proteins being partially or fully degraded. Cell envelope proteins incubated in E2, E1, M1 and M4 brines were more affected by UV irradiation, leading to 15–20% degradation. However, brine composition alone cannot account for the observed differences.

Interestingly, certain proteins were only identified by LC-MS/MS after irradiation (Fig. 3a-ii and b-ii). This suggests a role for UV photodissociation in protein degradation, thereby increasing the accessibility of certain proteins for enzymatic digestion prior to LC-MS/MS analysis. Finally, cytosolic proteins remaining even after extensive purification of the isolated cell envelope fractions prior to UV exposure displayed an increased sensitivity (2-3x compared to controls) to UV radiation-induced degradation (Fig. 3b-i). This suggests that, due to their position inside the membrane, cell envelope proteins were more protected from the irradiation effects.

To determine whether protein degradation was influenced by intrinsic UV sensitivity or by the specific brine environment, the number of hypersaline conditions in which each protein was degraded upon UV irradiation was analysed. It was observed that 51% of the 106 cell envelope proteins detected were degraded in only one specific brine, while only 3% were degraded regardless of the hypersaline condition. This underscores then the critical role of brine composition in modulating protein preservation under UV exposure. While the three proteins consistently degraded across all brine conditions were identified as an ABC-type transport system permease

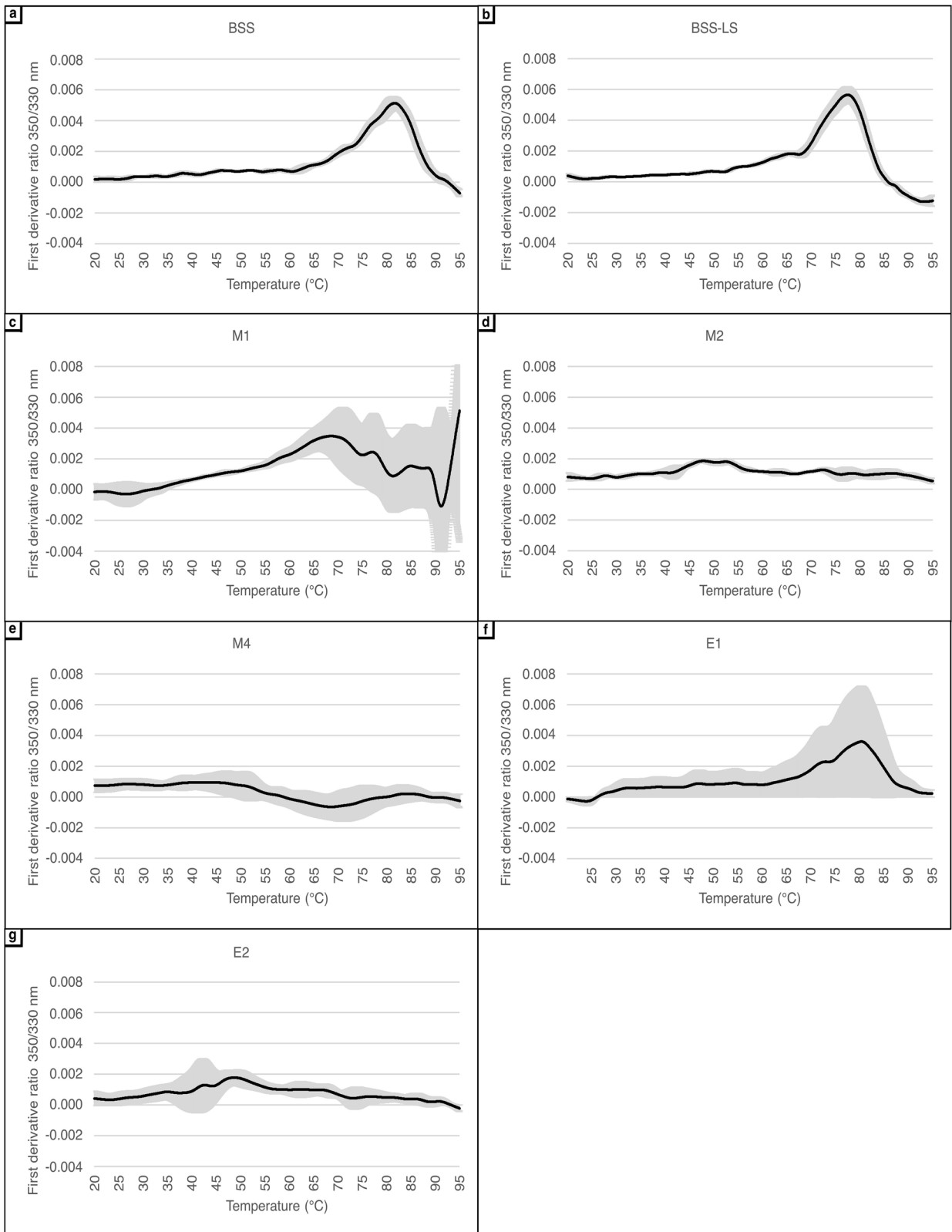

**Fig. 1 | NanoDSF first derivative of 350/300 nm ratio of cell envelope proteins in cell envelope extracts.** Temperature gradient was performed from 20 °C to 95 °C at a rate of 1 °C per minute. Grey areas represent the standard deviation calculated from five biological replicates. Panels (**a**–**g**) represent the different brines.

protein (Q9HMW2), a DUF1628 domain protein (Q9HNA7), and a $Na^+/H^+$ antiporter (Q9HNF7), no significant correlation between amino acid composition and protein degradation was found (Supplementary Fig. 3). Moreover, no common structural characteristic or positioning in the membrane were identified.

The results presented confirmed that brine composition directly affects the preservation of cell envelope biosignatures upon UV irradiation.

While chaotropicity played a significant role, it alone could not fully account for the observed differences in membrane protein degradation

## Table 2 | Relative changes in exposure of cell envelope protein Trp residues in different brines

| Brine | Emission Peak (nm) | Trp Exposure to Brine |
|---|---|---|
| M2 | 325 | -- |
| BSS (CTRL), E1 | 329 | ∅ |
| M1, BSS-LS, E2 | 331 | ∅ |
| M4 | 339 | +++ |

BSS is used as the reference point. A shift toward smaller wavelengths indicates more buried residues and a shift toward higher wavelengths more exposed residues. (−) Trp are more buried inside proteins; (∅) Trp are unaffected; (+) Trp are more exposed to the brine environment.

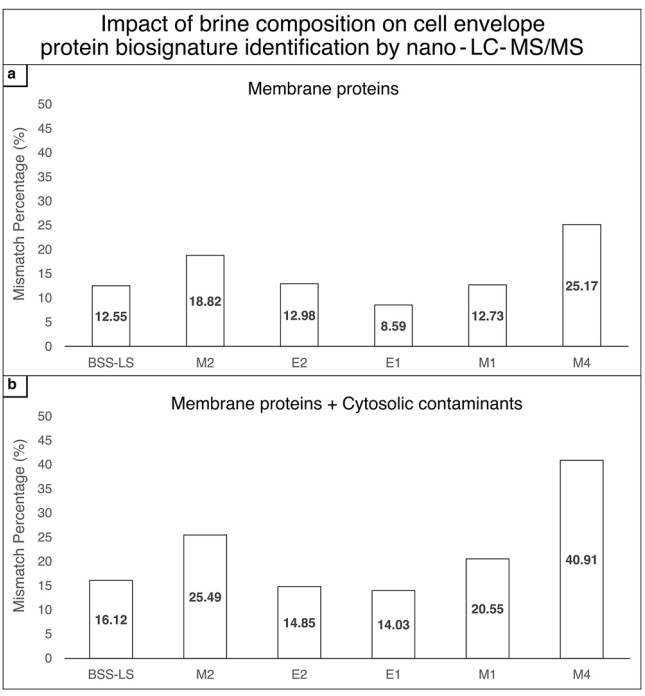

**Fig. 2 | Assessment of the effect of brines on cell envelope membrane protein composition through nano-LC/MS-MS.** The mismatch percentages represent the combined number of proteins that were either uniquely identified or not identified from cell envelope extracts incubated in the respective brines compared to those incubated in the control brine BSS. These mismatches resulted from chemical alterations in the cell envelopes after incubation in the various brines. Refer to the Materials and Methods for the discriminant parameters for protein identification. **a** Cell envelope proteins; **b** Cell envelope proteins + cytosolic contaminants in cell envelope extracts.

when irradiated. For example, brines M1 and M4, despite exhibiting opposite effects in terms of chaotropicity, caused similar levels of degradation after UV-irradiation. Consequently, a second effect, brine photochemistry, was investigated.

### Assessment of brine photochemistry on protein degradation

UV irradiation of the brines implies the formation of reactive species and radicals which lead to overall degradation of biomolecules. To investigate the photochemistry of the brines, general reactive oxygen species (ROS) production upon irradiation was evaluated in each solution using the fluorescent probe DHR123 (Fig. 4). Three groups could be observed: (1) M4, E2 and E1 brines had a high production of ROS (with M4 having a 16-fold increase fluorescence intensity compared to the next highest ROS-producing brine, E1), (2) M1 had a moderate production and (3) M2, BSS and M1 had a low production. No significant overall correlation between ROS production and proteins degradation was found (Spearman Correlation Test, *p*-values < 0.01). However, the group of brines producing

the lowest amount of ROS is the same group that exhibited better preservation of cell envelope proteins irradiated by UV (Fig. 3a-i).

The effects of UV irradiation on proteins can be directly measured through the generation of specific UV-induced chemical modifications via the production of reactive species and radicals (see Supplementary Table 1). The ratios of modifications in irradiated samples relative to non-irradiated control samples were compared using nano-LC-MS/MS data (Fig. 5, Supplementary Data 3). Low photochemical degradation, as indicated by ratios close to 1, were found as expected for cell envelope proteins incubated in BSS-LS and M2 (see Fig. 5a). However, for proteins in the BSS brine, a ratio approaching 3 was observed, suggesting a higher incidence of reactions with reactive species and radicals.

In contrast, for brines that induced higher protein degradation under UV irradiation (E1, E2, M1, and M4; Fig. 5b), most, especially M4 and E2, did not generate abundant photo-induced damages to cell envelope proteins upon UV radiation, with their respective ratios also close to 1. Although those brines showed high production of ROS, this phenomenon may be explained by potential extensive degradation of modified peptides, rendering them undetectable by mass spectrometry. Therefore, the number of photo-chemical modifications detected in the irradiated and control samples were similar, leading to ratios close to 1. The high ratio observed for M1 (> 4) could be then interpreted by revisiting Fig. 3a-i. Unlike M4, E2 and E1, M1 contained a higher proportion of partially degraded proteins compared to fully degraded ones, allowing for the detection of a greater number of modified peptides.

Like chaotropicity, photochemistry varied significantly across the different hypersaline conditions tested. While some parallels can be drawn between preservation properties and ROS production, this does not fully account for all the observed phenomena, particularly in the case of the M1 brine, which induces high protein degradation despite exhibiting lower oxidative activity.

### Acclimated cell envelope exhibits different preservation properties

The data presented thus far has primarily focused on the effects of brines and UV irradiation on *H. salinarum* cell envelope extracts. Given the hypothesis that cell envelopes derived from environmentally adapted microorganisms may exhibit different responses to these conditions, an acclimation process for *H. salinarum* to a selection of the brines was conducted.

Of all the brines attempted, successful acclimation was observed only when cells were cultured in the E1 medium. The brine has been reported as having a kosmotropic activity, therefore it was suggested that cell envelope acclimated in this brine would be more sensitive to chaotropic effect than the control cell envelope. In contrast, acclimation could not be achieved in the E2 and M2 brines, as cell death occurred during the process, likely due to large part to their chaotropic nature.

First, to determine if acclimation to different hypersaline conditions would induce compositional differences in cell envelopes, the lipid and protein compositions of E1 acclimated cell envelope were first investigated and compared to control (non-acclimated) cell envelope (Fig. 6). For cell envelope proteins, a majority (60%) of identified cell envelope proteins were shared between both conditions. However, a non-negligeable proportion of proteins (25%) were no longer identified after the acclimation to the E1 brine, whereas 15% of proteins were only recovered in the E1 brine (Fig. 6b). The proteins identified in each condition are listed in Supplementary Table 2. An interesting difference between the two conditions was the identification of bacteriorhodopsin (bR) in the acclimated samples. bR is a light-driven proton pump capable of directly absorbing UV-C radiation, which may increase the overall UV sensitivity of the cell envelope.

Four major groups of *Halobacterium* membrane lipids were determined to be most affected by acclimation of cells to the E1 brine (Fig. 6a). A significant difference between the conditions was observed in the proportion of Phosphatidylethanolamine-Diphytanyl glycerol diether (PE-DGD), which was substantially present in acclimated samples, whereas it was nearly undetectable in control (optimal *Halobacterium* growth medium) samples.

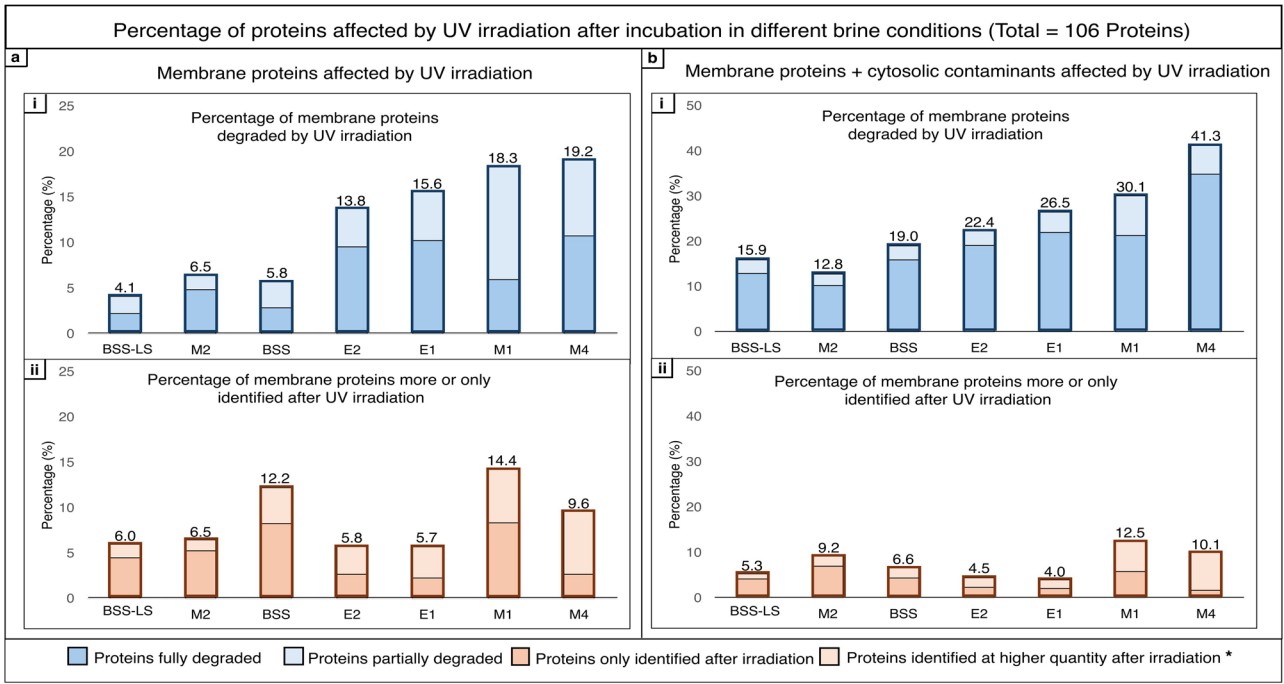

**Fig. 3 | Assessment of brine preservation properties under UV irradiation.** Cell envelope extracts were incubated for 65 h before UV-irradiation to 28.2 kJ.m$^{-2}$ (185–1000 nm) prior to nano-LC-MS/MS analysis and quantification. Five biological replicates were analysed per condition, and presence of proteins in each condition was validated if present in 3/5 replicates and if at least two significant peptides were identified. **a** Cell envelope proteins. **b** Cell envelope proteins + cytosolic contaminants. (i) Percentage of proteins fully or partially degraded by UV irradiation. (ii). Percentage of proteins only or more identified in irradiated samples. *Proteins found in the irradiated sample with a 2-fold change > 2 compared to non-irradiated sample.

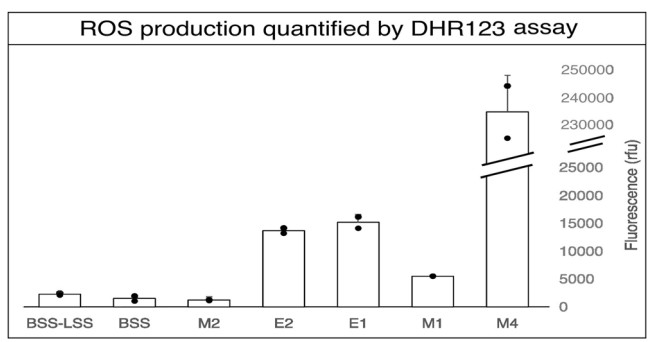

**Fig. 4 | Photochemistry of the different brines assessed by general ROS production using the fluorescent probe DHR123.** Four different samples were assessed for each brine: the non-irradiated brine without DHR123, the irradiated brine without DHR123, the non-irradiated brine with DHR123 and the irradiated brine with DHR123. Triplicates of each sample were pipetted into a microwell plate. Fluorescence was measured at 520 nm after excitation at 480 nm.

PE-DGD plays a crucial role in maintaining membrane integrity and fluidity, both of which are essential for cells undergoing acclimation processes. Additionally, more Extended Phosphatidylglycerol Diphytanyl glycerol diether (PG-Ext DGD) and Extended phosphatidylglycerophosphate-methyl ester Diphytanyl glycerol diether (PGPMe-Ext DGD) were also found in acclimated cell envelope, the latter being a known component of bR purple membrane. Conversely, less Mono/Di-unsaturated Phosphatidylglycerophosphate- Diphytanyl glycerol diether (PGP-DGD insat) and SO$_3$-Tetraglycolipid- Diphytanyl glycerol diether (SO$_3$-Hex4-DGD) were found in the cell envelopes of E1-acclimated cells.

Secondly, to assess if proteins and lipids compositional changes induced by the acclimation to brines of differing kosmotropicity would influence cell envelope fragments preservation from UV radiation, both control (BSS) and acclimated (E1) cell envelope extracts were incubated in a chaotropic brine (terrestrial brine E2) and exposed to UV radiation (28.2 kJ.m$^{-2}$). Acclimated cell envelope extracts were found highly susceptible to UV irradiation degradation compared to non-acclimated samples, as only two out of four pellets were recovered post-irradiation following ultracentrifugation recovery. This was confirmed by LC-MS/MS analysis by performing proteins label-free quantification of samples before and after irradiation (Fig. 7). Compared to the results presented in Fig. 3, cell envelopes incubated in the E2 brine also exhibited extensive degradation. In the control cell envelope, 32% of proteins were affected by both brine incubation and UV treatment, whereas 44% were impacted in the acclimated cell envelope. Additionally, 23% of cell envelope proteins were degraded in the control samples, while 38% were degraded in the acclimated cell envelope samples. These findings confirm the idea that the lipid and protein compositions of cell envelope fragments are closely linked to their preservation potential from both chaotropicity effects and UV irradiation.

## Discussion

The ability of salts to preserve biosignatures over geological timescales has recently gained significant interest as putative ancient biosignatures are found in halite and other evaporites. However, brine preservation capacity has remained largely speculative, due to sparse experimental data attributable to challenges in analysing hypersaline solutions. Therefore, this study has aimed to provide insights into the effects of brine ion composition on the preservation of microbial cell lysis-associated biomolecular fragments. The extreme halophile *Halobacterium salinarum* was selected for these experiments due the documented evidence of its entrapment in ancient halite fluid inclusions[5], the availability of accessible data on membrane proteins, and a well-characterized cell envelope. Brines of different compositions used in this work are analogues of site locations on Mars: the alkaline-carbonate-chloride dominated M1-brine from Gale Crater-Nakhla meteorite, the magnesium-sulphate-chloride dominated M2 brine from Meridiani Planum and the magnesium perchlorate M4 brine from the Mars North Pole.

As previously mentioned, incubation of *H. salinarum* cell envelopes in the acidic iron-rich M3 brine resulted in an absence of extractable

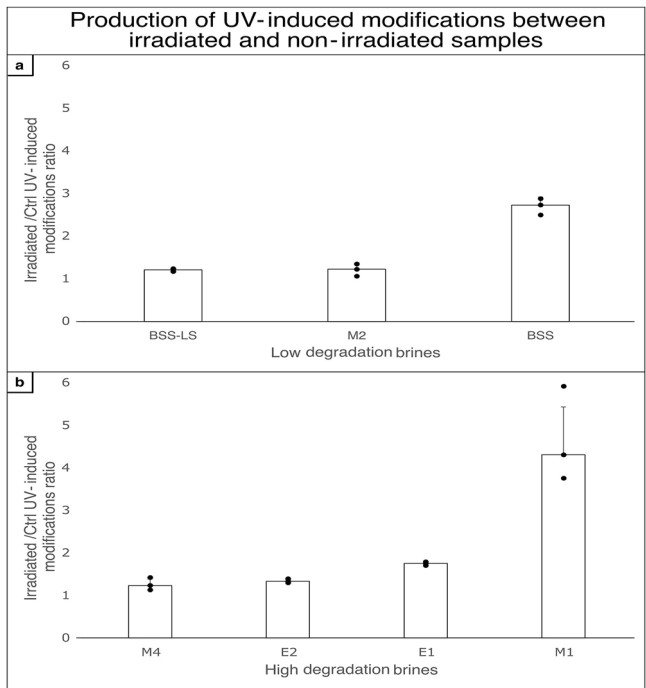

**Fig. 5 | Photochemistry of the different brines assessed by the identification of specific photo-chemical modifications in the irradiated samples. a** Brines allowing for low degradation of cell envelope proteins upon UV radiation. **b** Brines allowing for high degradation of cell envelope proteins upon UV radiation.

membrane proteins. Instead, a brown precipitate formed during incubation, likely due to the high $Fe^{2+}$ content of the brine. This can lead to ionic interactions with the negatively charged surfaces of S-layer proteins resulting in mineral formation in a process known to occur in the presence of excess sulphates, carbonates, or phosphates[23,26,49,50].

Also included were analogues of hypersaline environments from different geological periods on Earth: the chloride dominated E1 brine from the Guadalupian era, the chloride-Magnesium dominated E2 brine from the Eocene era and the chloride dominated BSS brine from modern Earth. As the surface of Mars and other planetary bodies hosting hypersaline environments are continually exposed to solar radiation, the effects of full spectrum UV radiation on the preservation properties were assessed. For Earth environments, UV radiation is not of direct relevance due to the protective properties of the ozone layer. However, the use of ancient terrestrial brine analogues (E1, E2) and a modern terrestrial halophilic microorganism (*H. salinarum*) provides a reference point for preservation of biosignatures for life as we know it, forming the basis for a more complete understanding of the complex interplay of biochemistry, brine composition and UV photochemistry in biosignature preservation.

The results presented here demonstrate that variations in brine ion composition significantly affect the preservation potential of cell envelope proteins, with and without UV irradiation.

Firstly, the brine composition dictates the disordering or ordering of macromolecular structures by altering interactions with water molecules[35], referred to respectively as chaotropicity and kosmotropy. The traditional agar gelation point assay used in this study provided a good assessment of the chao-kosmotropic nature of the different brines on a simple macromolecular system (agar). However, some differences were observed with previously reported data for some of the brines used here[14] which were attributed to differences in precise type of agar and instrument used, revealing the acute sensitivity of this methods when working with mixed brines. This theoretical chao-kosmotropic classification based on brine interactions with agar was then confirmed by Nano-DSF, which allowed the investigation of these effects directly on the cell envelope macromolecules

(protein Trp residues) in non-diluted brines. Relative to the control BSS, brines M1 and E1 exerted a more kosmotropic activity while the magnesium containing brines, M2, M4, and E2 increased structural disorder of cell envelope extracts due to their more chaotropic nature. The latter may be attributed to the higher concentrations of $Mg^{2+}$ ions present in these brines, which are noted as chaotropes[36].

Measurements of tryptophan residues fluorescence spectra after incubation in the structural destabilizing brines brought nuance to their chaotropic activity by focusing more precisely on their effect on membrane protein side chain positioning. The Trp emission spectra of cell envelopes incubated in the M2 brine corresponded to a greater number of Trp residues solvated within the lipid membrane structure compared to incubation in the control BSS brine. In contrast, Trp residues of cell envelope proteins incubated in the M4 brine became more exposed to the brine environment. Finally, in the E2 brine, no apparent effect was detected. These results highlight the complexity of chaotropicity, at the macromolecular and molecular levels. An additional indication of structural changes in the cell envelope was demonstrated by the variation in protein identification by mass spectrometry across the different brines. This was particularly evident in the M2 and M4 brines, which exhibited important differences in number of identified proteins (≃20%) compared to the control salt (BSS). This underscores that the structural alterations induced by the brine environment can interfere with analytical chemistry workflows, particularly for mass spectrometry, a technique commonly used not only in laboratories but also in spacecraft and rovers for biosignature detection.

The structural changes induced by brine composition could also alter the susceptibility of the cell envelope proteins exposed to UV-radiation by modifying accessibility of UV-absorbing protein regions[51].

Increased exposure of UV chromophoric amino acids (Trp, Tyr, Phe, His, Cys)[52] to UV-irradiation can lead to a series of additional protein damages resulting from photo-oxidation of side changes causing structural modifications and/or fragmentation of protein side chains[53], which can internally lead to increased peptide bond UV exposure and subsequent dipeptide $C_\alpha$-C bond breakage[54]. Thus, the increased Trp exposure and protein degradation observed for cell envelope extracts incubated in the chaotropic M2 and M4 brines may also result in increased sensitivity to UV degradation. It also highlights the complex interplay of brine chaotropicity and UV photochemistry on biosignature preservation.

Analyses of brine photochemistry revealed that it can be the predominant effect in cell envelope protein degradation over chao-kosmotropicity. This was supported by investigating the production of radicals and reactive species in the absence of any biological material using a ROS sensitive probe (DHR 123), as well as the effects on cell envelope biosignatures revealed by the proportion of photo-induced modifications in irradiated and non-irradiated samples. Although detection of such modifications in irradiated samples was limited due to extensive degradation of modified peptides, photochemistry was central to explain the preservation patterns of the brines.

A generalized model for the effects of brine UV-photochemistry on cell envelope biosignature preservation can be proposed. Upon exposure to UV radiations, particularly in the UV-B and UV-C range, photochemical reactions occur, resulting in the generation of specific reactive species and radicals (see Supplementary Table 3)[55–61] known to cause important damages to the cell envelope integrity by directly reacting with proteins and lipids. Protein Trp, Tyr, Cys, and Met residues are more susceptible to radicals[62], leading to direct fragmentation of the proteins and/or structural changes. Lipid peroxidation leads to the formation of lipid hydro-peroxides that can subsequently react with proteins[63,64].

Hydroxyl radical (•OH) and hypochlorite acid are two potent oxidants generated upon UV irradiation in all the brines present in this study. They lead to extensive protein damage through the formation of carbon-centred radicals that can further react with dissolved oxygen, producing peroxy radicals leading to auto-fragmentation if formed at the α-carbon[48,57]. Hypochlorous acid is also capable of cleaving peptide bonds and oxidizing amino acids[48,65,66] as well as leading to chlorination of tyrosine which induces

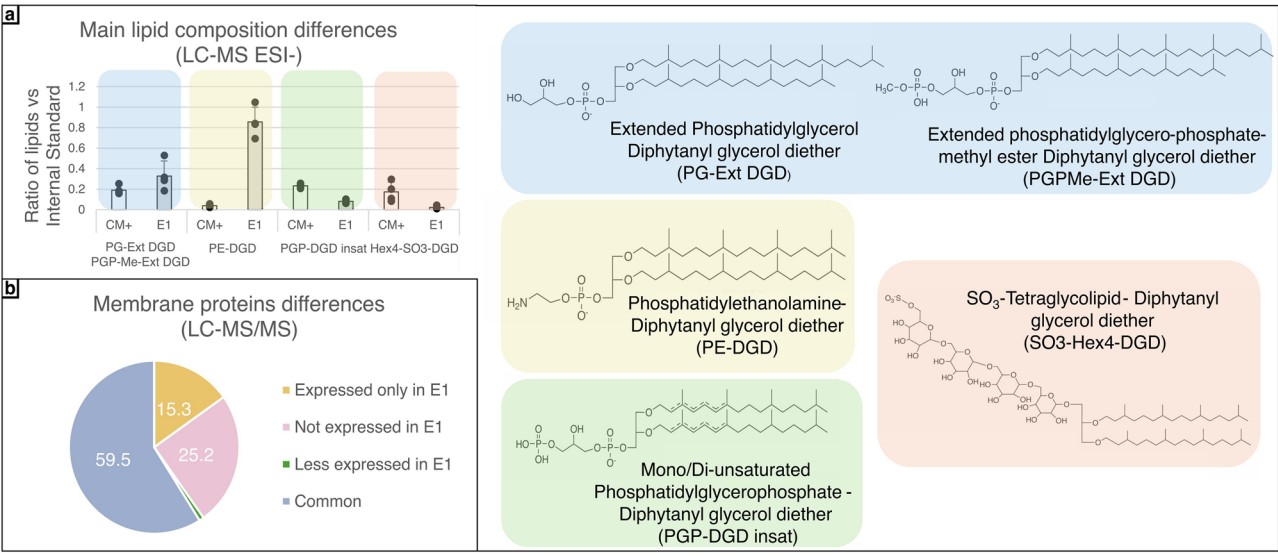

**Fig. 6 | Compositional differences between acclimated (E1) and control (CM + ) cell envelope. a** Main differences in lipid composition between acclimated and control cell envelopes. Lipids were identified by LC-MS and analysed using DataAnalysis (Bruker). Semi quantification was performed using 1,2-Distearoyl-sn-glycero-3-phospho-rac-(1-glycerol) as internal standard. **b** Comparison of cell envelope protein composition between acclimated and control cell envelopes. Proteins were analysed by mass spectrometry and treated using PeaksX Studio.

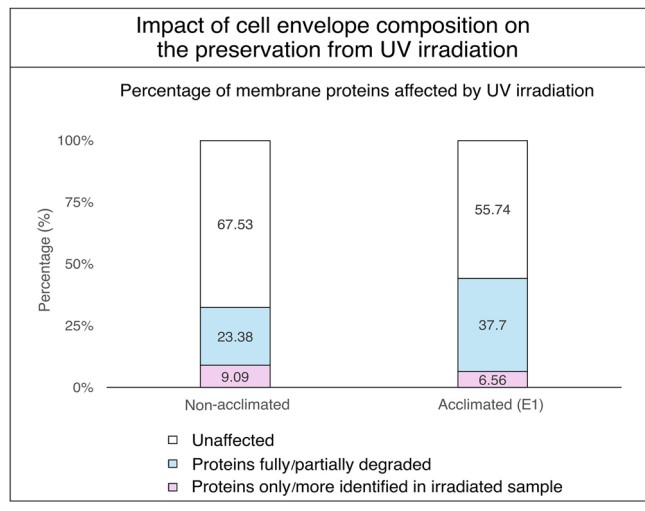

**Fig. 7 | Acclimation experiments.** Comparison of UV sensitivity of cell envelope proteins between cell envelopes from acclimated to brine E1 and control *Halobacterium* cell cultures grown in standard CM+ growth medium. Cell envelope proteins were analysed by mass spectrometry and label-free quantified using PeaksX-Pro software.

large-scale protein self-organisation perturbation[67]. These observations may explain the difference in preservation efficacy between the chloride-dominated brines BSS and BSS-LS. Both conditions demonstrated effective preservation properties, however, BSS-LS exhibited slightly superior performance. This improvement may be attributed to the lower chloride ion content in the BSS-LS brine.

Moreover, bicarbonate/carbonate anions ($HCO_3^-/CO_3^{2-}$), uniquely present in the M1 brine, have been described in ref. 68 as hydroxyl radical scavengers by forming carbonate radical anions ($CO_3^{\bullet-}$) and have a high reactivity with specific amino acids[69]. This scavenging activity might explain why the M1 brine exhibits a reduced ROS production although DHR123 reacts with carbonate radicals, but at a slower rate[70,71]. However, it has been reported that in carbonate–bicarbonate containing systems, many

hydroxyl-radical-induced processes may be carried out through carbonate radicals instead[63,72]. This phenomenon could provide an explanation for the significant degradation of cell envelope proteins observed in such systems.

Additionally, it is not surprising that the magnesium perchlorate brine M4 exhibits significant degradation of cell envelope proteins as perchlorate is well-known for its great oxidative properties, which lead to organic matter degradation[73].

Finally, sulphate radicals $SO4^{\bullet-}$, presumably abundant in the M2 brine, also lead to extensive oxidation of proteins as it has been reported that it has a longer half-life and oxidation than $\bullet OH$[74]. However, reactive species and radical production also depends on physio-chemical parameters like pH[75]. The production of $\bullet OH$ is reduced in acidic environments, which may explain the limited detection of ROS in the M2 brine (pH = 1). Additionally, certain oxidation reactions are less prevalent under acidic conditions.

Although reactive species and radicals have an important impact on the degradation of proteins and lipids, they are not solely responsible as no significant correlation was found between ROS production and protein degradation. Chao-kosmotropic effects discussed previously may render the cell envelope proteins and lipids more accessible to ROS/radicals-related degradation.

Biosignature preservation in brines is thus a complex interplay between ionic composition, UV-photochemistry, and the biological macromolecules. The dominant factor(s) influencing preservation properties vary between brines. In some cases, photochemical processes appear to be the primary influence, as observed in the comparison of E1 versus E2 brines, as well as BSS versus BSS-LS versus M2. In both instances, brines exhibited similar intra-group protein preservation properties and produced comparable levels of ROS, despite having divergent (chao-kosmotropic) effects on macromolecular structures. Thus, chaotropicity does not appear to be the most important factor in the degradation of cell envelope proteins in these cases. Moreover, the kosmotropic brines E1 and M1, despite promoting a more structured organisation of proteins and lipids, did not provide a protective effect, as both exhibited significant degradation of proteins under UV exposure. Conversely, substantial differences in ROS production observed between the E1 and M4 brines can lead to similar levels of membrane protein degradation during UV irradiation. These similarities may be explained by differences in chaotropic activity or other effects not investigated in this study.

Overall, for early terrestrial analogues, both Guadalupian (E1) and Eocene (E2) brines demonstrated comparable levels of ROS production, yet exhibited contrasting chaotropic activities. Despite these differences, both brines induced similar degradation of cell envelope proteins upon exposure to UV irradiation. Moreover, compared to modern Earth Great Salt Lake brine (BSS), preservation was found more pronounced in the early Earth brines (E1, E2). In the Martian context, Merridiani Planum (M2) brine displayed notable preservation potential, characterized by reduced ROS production, although it exhibited chaotropic effects on cell envelope proteins. Inversely, the perchlorate-dominated Martian north polar brine simulant (M4) showed high photochemical reactivity due to its perchlorate composition, leading to significant degradation of organic material. Similar conclusions were made for the Gale Crater/Nakhla meteorite-carbonate-related environment (M1 brine) although it displayed kosmotropic activities. These findings suggest once more that while photochemistry does not account for all the observed differences in preservation, it can play a dominant role in influencing the long-term stability of organic matter in these environments. Furthermore, this screening has now provided a framework for investigating sites where biosignature could be preserved in fluid inclusions. On Mars, the composition of brines found in the Meridiani Planum region has emerged as having interesting preservation properties, while the Martian North Pole and Gale Crater have not demonstrated similarly favourable conditions. On Earth, given that the ozone layer effectively blocks UV-C radiation, both selected sites show promise. However, the Guadeloupian San Andres site is preferred, as it does not possess a chaotropic activity, which may enhance the preservation of biosignatures.

While this study provides new insights into the preservation of cell envelope proteins within fluid inclusions under full spectrum UV irradiation, the complexity of these systems must be acknowledged. The contents of lysed microorganism-filled fluid inclusions are not limited to cell envelope fragments, but also include the full range of cytosolic components. In the case of *H. salinarum*, this would result in the significant release of DNA into the environment, thereby increasing the overall viscosity. Increased viscosity can slow chemical reaction rates, potentially reducing the rate of detrimental oxidation reactions. Moreover, the release of additional molecules and proteins would also increase the number of chromophore sites available for interaction with UV light, thereby raising the likelihood of side reactions that would degrade the cell envelope extracts. On the other hand, in natural environments, fluid inclusions would be further protected by the surrounding crystal matrix, which would reduce the total irradiance received by the system and provide greater protection than the experimental setup used here. Moreover, anoxic conditions and gas composition found on Mars were not addressed in this study. Finally, based on that the acclimation experiments conducted, it is clear that alterations in the composition of cell envelope lipids and proteins significantly influence the overall resistance of fragments to UV irradiation. This underscores the necessity of further investigating the resistance of the cell envelopes of different microorganisms to irradiation to better understand their potential for preservation under such conditions.

Moreover, this study also highlights the complexity of applying NanoDSF to highly intricate samples and salt conditions, far exceeding the simplicity of traditional setups. While the technique offered valuable insights, its limitations necessitated complementary analyses to interpret the observed phenomena. Furthermore, the observed transitions could be influenced by a small number of highly expressed proteins affecting the overall fluorescence response, as previously demonstrated[76]. Given that transmembrane transducer proteins and S-layer glycoproteins were highly abundant in the cell envelope extract, NanoDSF provides valuable insights. However, the information obtained may be incomplete. Future research should prioritize integrating alternative methods to expand upon these findings, providing a more comprehensive picture of the chaotropic and kosmotropic effects on cell envelope stability, particularly in challenging brine conditions.

Finally, as the preservation properties of brines represent a complex and not yet fully understood phenomenon, with chaotropicity and photochemistry interacting, it is essential to conduct precise analogue experiments prior to the selection of sample recovery sites. These experiments, following the framework established here, should focus on evaluating the preservation potential of these environments, with particular attention to photochemical processes.

## Methods

### *Halobacterium salinarum* NRC-1 culture

*Halobacterium salinarum* NRC-1 strain was obtained from Dr. Caryn Evilia (Idaho State University). Five biological replicates of *Halobacterium salinarum* NRC-1 were incubated from pre-cultures in Complex Medium+ (CM + : 4.28 M NaCl, 81 mM $MgSO_4.7H_2O$, 27 mM KCl, 10 mM trisodium-citrate.$2H_2O$, 1% (w/v) peptone Oxoid® LP0034, 0.5% (v/v) 100% Glycerol, Metal trace solution: 6.3E-6 M $FeSO_4.7H_2O$, 1.5E-6 M $ZnSO_4.7H_2O$, 2.19E-6 M $MnSO_4$, 4E-8 M $CuSO_4.5H_2O$, pH adjusted to 7.4) at 42 °C 220 rpm in the dark until they reached stationary phase ($OD_{600}$ = 1.3-1.4).

### Cell envelope extraction

Cultures were harvested by centrifugation at 10,000 x g for 10 min and the cell pellet resuspended in 10 mL of Basal Salt Solution (BSS: CM+ without organics). Cell lysis was achieved by a single freeze/thaw cycle using liquid nitrogen. As *H. salinarum* cells are highly polyploid, each sample was treated with 4 mg of DNAse 1 (DN25, Merck) for 30 min at 37 °C using a Tube Revolver (force 15).

Raw cell envelope fractions were harvested by centrifugation 20,000 x g for 2 h at 4 °C and washed using three successive ultracentrifugation cycles (70.1Ti Rotor, Beckman) at 100,000 x g 30 min 4 °C with 10 mL of fresh BSS per cycle. The resulting cell envelope pellets were then resuspended in 1 mL 0.1 M Tris-HCl buffer, pH 7.4 (Tris buffer) and any remaining cytosolic contaminants removed using a 20–55% sucrose gradient (5% increments in Tris buffer) with centrifugation at 80,000 x g 15 h at 10 °C. The red, carotenoid-bearing cell envelope fractions were collected and washed three times in Tris buffer by centrifugation at 229,600 x g 4 °C for 1 h. The purified cell envelope fractions were resuspended in 2 mL of Tris buffer using an ice-cold ultrasonic bath (Advantage Lab, AL-04-04) for 5 min. Proteins were quantified by bicinchoninic acid assay (BCA) (Pierce™ BCA Protein Assay Kit, ThermoFisher Scientific), with 125 μg aliquots of cell envelope lyophilized and stored at −20 °C until use.

### Brine solution preparation

The brines used in this study were selected to mimic various Early Earth and Mars environments (see Table 3)[77–79]. The pH of each solution was not adjusted to preserve conditions more closely resembling the natural environment being modelled. As the brines were supersaturated, the solutions were prepared in borosilicate bottles by first adding the required salts to ultrapure (MilliQ®) water ($MQH_2O$) at approximately 20% of the final desired solution volume. The mixture was continuously stirred with a magnetic stirrer for 1 h, then the volume completed with $MQH_2O$ before incubation for 5 days at 30 °C to equilibrate the liquid and solid phases. Finally, each bottle was sealed with three layers of parafilm to avoid evaporation and stored in the dark at room temperature. The ionic composition of brine M3 was found to be incompatible with most methods used in this study in ways that no conclusion could be drawn for this brine.

### Incubation of cell envelope in brine solutions

Cell envelope aliquots (125 μg) were incubated in 500 μL of each brine in 1.5 mL microcentrifuge tubes using an ice-cold ultrasonic bath to homogenize the solutions. The tubes were sealed with parafilm and incubated in the dark at 18 °C for 65 h.

### UV-Irradiation

A solar simulator was used for total UV exposure (Xe Lamp Hamamatsu L2479, 185-1000 nm) to better compare potential biosignature preservation

**Table 3 | Brine compositions**

| Simulated Fluid inclusion brine composition (mol/L) | Early Mars | | | | Early Earth | | Controls | |
|---|---|---|---|---|---|---|---|---|
| | Type I (Nakhla Martian meteorite) | Type II (Merridiani plunum) | Type III (Same as II but more acidic) | Type IV (Phoenix Lander site North Pole) | San Andres Formation (Guadaluoian, 274–272 Ma) | Bresse Bassin (Eocene, 36-34 Ma) | | |
| | M1 | M2 | M3 | M4 | E1 | E2 | BSS | BSS-LS (low salt) |
| NaCl | 1.27 | 2.27 | 1.36 | | 5.56 | 2.66 | 4.28 | 2.89 |
| KCl | 3.78 | 1.33 | 1.14 | | 0.27 | 0.53 | 0.27 | 0.27 |
| MgCl$_2$.6H$_2$0 | | 1.15 | 3.70 | | 0.18 | 1.33 | | |
| MgSO$_4$.7H$_2$0 | | 2.55 | | | | | 0.81 | 0.81 |
| HCl | | 0.39 | 0.11 | | | | | |
| tri-Sodium Citrate.2H$_2$O | | | | | | | 0.13 | 0.13 |
| CaCl$_2$ | | | | | | 0.60 | | |
| KHCO$_3$ | 2.24 | | | | | | | |
| FeSO$_4$.7H$_2$O | | | 2.31 | | | | | |
| FeCl$_2$.4H$_2$O | | | 0.99 | | | | | |
| Na$_2$SO$_4$ | | | | | 0.80 | | | |
| Mg(ClO$_4$)$_2$ | | | | 4.16 | | | | |
| pH | 8.3 | 1 | 0.5 | 4.6 | 4.68 | 5.67 | 7.4 | 7.4 |

in representative early Earth and early Mars brines. The spectrum was recorded, and total irradiance was calculated (see Supplementary Fig. 4). *H. salinarum* cell envelope extracts incubated in the different brine solutions were exposed to 28.2 kJ.m$^{-2}$ of UV radiation (200–400 nm), based on preliminary exposure assays (see Supplementary Fig. 5), and compared to non-irradiated control samples. For this, samples were first diluted in the corresponding brine solution to a total volume of 8 mL and placed in a glass crystallizer (7 cm diameter), itself placed on 3D orbital shaker positioned underneath the solar simulator lamp. Both irradiated and control cell envelope fragments were collected for analyses. As all brines (except BSS and BSS-LS) were too dense for adequate separation, all samples were diluted with an equal volume of MQH$_2$O. Following ultracentrifugation for 2 h at 229,600 x g 4 °C, the recovered pellets were resuspended in 100 µL Tris buffer and fully resuspend using an ice-cold ultrasonic bath for 5 min.

### Mass spectrometry analyses of cell envelope proteins
**Cell envelope protein isolation.** Cell envelope proteins were isolated from cell envelope extracts following the previously established protocol[80] Briefly, 0.5 mL of methanol was added to each sample, mixing by vortex for 10 s followed by centrifugation for 10 s at 9000 x g. Subsequently, 0.25 mL of chloroform was added, with vortex mixing and centrifugation as previously. MQH$_2$O (0.375 mL) was added, with vortex mixing for 20 s and centrifugation for 1 min at 9000 x g. The upper methanol phase was discarded, and 0.375 mL of methanol was added to the remaining chloroform and protein-bearing inter- phases. Samples were mixed by inversion 20 times then by vortex for 20 s, followed by centrifugation for 5 min at 9000 x g. The lipid-bearing supernatant was recovered, and the protein pellets were air-dried on ice for 10 min. Both protein pellets and lipid fractions were stored at −20 °C until further use.

**Tryptic digestion.** Protein pellets were resuspended in 10 mM Tris buffer (pH 7.4) containing 0.3 mg.µL$^{-1}$ RapiGest™ SF (Waters) and subjected to bath sonication to homogenize the solution. Proteins were then reduced with 10 mM dithiothreitol and incubated on a Thermo-Mixer® from Eppendorf (1 h at 57 °C, 950 rpm 2D Orbital shaking). Subsequently, samples were alkylated with 15 mM iodoacetamide and incubated for 1 h at 20 °C in the dark at 950 rpm. Finally, proteins were digested using Trypsin Gold (Promega). A 1:20 (Enzyme:Protein) ratio was applied and samples were incubated for 15 h at 37 °C 450 rpm.

Digestion was terminated by addition of 1% trifluoroacetic acid and samples were desalted by solid phase extraction using SEP-PAK Plus C18 column (Waters). SEP Columns were conditioned by successive addition of 10 mL of methanol, acetonitrile (ACN) and 0.1% formic acid (FA). Samples were diluted in 5 mL of 0.1% FA and loaded onto the column. The column was washed once with 10 mL of 0.1% FA and peptides were recovered by two successive elutions with 1 mL of 80% ACN. Peptides were dried using a SpeedVac and resuspended to 1 µg/µL in 20% ACN just prior to LC-MS/MS analysis.

**Nano-LC-MS/MS orbitrap.** Peptide digests (1 µg) were injected and the samples were concentrated on a C18 cartridge (Dionex Acclaim Pep-Map100, 5 µm, 300 µm i.d. x 5 mm) and eluted on a capillary reverse-phase column (nanoEase M/Z Peptide CSH C18 Column, 130 Å, 1.7 µm, 75 µm X 250 mm) at 220 nL/min, with a gradient of 2% to 40% of buffer B in 45 min (A: 0.1% aq. FA/ACN 98:2 (v/v); B: 0.1% aq. FA/ACN 10:90 (v/v)), coupled to a quadrupole-Orbitrap mass spectrometer (Q Exactive HF, ThermoFisher Scientific) using a Top 20 data-dependent acquisition MS experiment: 1 survey MS scan (400–2.000 m/z; resolution 70,000) followed by 20 MS/MS scans on the 20 most intense precursors (dynamic exclusion of 30 s, resolution 17,500).

### Cell envelope structural stability analyses
For all molecular stability analyses, cell envelope extracts were incubated at 0.5 µg/mL for 65 h.

**Nano Differential Scanning Fluorometry (NanoDSF).** NanoDSF was used to monitor membrane protein folding/unfolding processes across a temperature gradient by tracking the intrinsic fluorescence of tryptophan residues (Trp) with excitation at 280 nm and emission measured at 330 nm and 350 nm. This method has previously been successfully applied to evaluate the effects of salts on soluble (cytosolic) proteins from *H. salinarum*, but not the membrane proteome[36].

Brine solutions were placed in quartz capillaries and subjected to a temperature gradient from 20 °C to 95 °C (1 °C/min). Following the detection of crystallisation during the preliminary sample runs, sealing glue was applied to capillary ends for all subsequent experiments.

**Tryptophan Fluorescence.** Shifts in tryptophan fluorescence were monitored using a spectrofluoremeter (Jasco, FP-8550) by excitation at

295 nm, measuring changes in emission from 300 to 405 nm based on the degree to which Trp residues are either buried within proteins or exposed to the solvent[81].

## Chaotropicity agar gel-point measurements

Brine chaotropicity was measured using the agar gel-point method[33]. 500 μL aliquots of each brine were prepared in 1.5 mL microcentrifuge tubes to 1:3 ratio (MQH₂O:Brine). The sample tubes were then pre-warmed for 30 min in an oven at 60 °C together with all required materials (micropipette and tips, 96-well microplate), and a 3% agar (Naclai tesque, 01056-15) solution made in MQH₂O. Each of the pre-warmed brine solutions was mixed 1:1 (v/v) with 500 μL of 3% agar solution. 200 μL aliquots of each mix were added in triplicate to the 96-well microplate. The plate was immediately inserted into a FLUOStar® (Omega-BMG Labtech) microplate reader pre-heated to 45 °C, and the plate equilibrated to 45 °C for 1 h prior to commencing measuring absorbance at 500 nm over a series of decreasing temperatures until agar gelation was detected for all test solutions when absorbance started increasing. Temperature was decreased in 0.5 °C increments and absorbance recorded after a 5 min incubation.

## Reactive oxygen species production in brines after UV-irradiation

A reactive oxygen species (ROS)-activated fluorescent probe, Dihydrorhodamine 123 (DHR123), was used to assess the effects of ionic composition on brine UV photochemistry. Each brine was assayed with 0.5 μM DHR123 after exposure to 3 min of UV radiation (28.2 kJ.m$^{-2}$). Samples were then incubated in a ThermoMixer® (Eppendorf) 5 min at 37 °C, 950 rpm in the dark before measuring the fluorescence emission at 520 nm after excitation at 480 nm.

## Acclimation of *H. salinarum* cultures to Early Earth/Early Mars brines

The cell envelope extracts tested previously were extracted from *H. salinarum* cultures prepared in optimal, modern Earth brine conditions, rather than the Early Earth/Early Mars brines considered in the study. To test the effects of differential protein expression due to cell acclimation on cell envelope protein preservation as biosignatures, cultures of *H. salinarum* were first acclimated to the E1, E2 and M2 brines using a stepwise process (Supplementary Table 4). Four biological replicates were cultivated in CM+ growth media until they reached exponential phase (as determined by growth curves for each acclimation step). Cultures were then diluted to $OD_{600} = 0.05$ in the proceeding step for each of the three growth media types (E1, E2, M2), following the four-step protocol, and incubated as for the other experiments described in this study. Growth was monitored by both $OD_{600}$ and phenotypic variations by optical microscopy. When stress was observed (reduced generation time, cell morphology changes), several dilution/growth cycles were repeated in the same media type to allow proper acclimation of the cultures. Once full acclimation in desired medium (after all four acclimation steps) was achieved, 50 mL cultures were grown to stationary phase ($OD_{600} = 1.3–1.4$). Cell pellets were then harvested for cell envelope extraction as for the other experiments.

To determine whether compositional changes in cell envelope fragments could influence their UV irradiation preservation properties, both control and acclimated samples were incubated in a third, unrelated chaotropic brine, rather than the brine used for acclimation. This strategy was employed to assess whether compositional change would influence chaotropicity resistance as well as to create conditions more likely to reveal differences in preservation upon UV irradiation. The irradiation of cell envelope extracts as well as the extraction of cell envelope proteins and lipids were performed in the same manner as for the other irradiation experiments.

## Evaluation of the influence of cell envelope composition on biosignature preservation.

Differences in both membrane protein composition and lipid composition between control and acclimated cell envelopes were assessed by mass spectrometry, with the goal of assessing how compositional changes affected biosignature preservation in brines with and without UV-irradiation.

Protein digests were analysed by ultra-high-performance liquid chromatography–tandem mass spectrometry (UHPLC-MS/MS) on an Ultimate 3000- RSLC system (Thermo Scientific) connected to an electrospray ionisation–quadrupole–time of flight (ESI-Q-TOF) instrument (Maxis II ETD, Bruker Daltonics). The separation was achieved using a RSLC Polar Advantage II Acclaim column (2.2 μm, 120 Å, 2.1 × 100 mm, Thermo Scientific) with the following linear gradient using mobile phase A (MQH₂O + FA 0.1%) and B (LC-MS grade ACN + FA 0.08%), at 300 μL/min: increased from 2% B to 37% B for 30 min, and to 60% B in 5 min, increased to 80% B in 2 min, increased to 100% B in 2 min and hold for another 2 min. Next, decreased to 2% B in 2 min and equilibration for 2 min (total run time 43 min). The MS spectra were recorded in positive ion mode in the m/z range 150–1800. The source parameters were as follows: nebulizer gas 35 psi, dry gas 8 L/min, capillary voltage 3500 V, end plate offset 500 V, temperature 200 °C. Analyses were performed using collision-induced dissociation in data dependent auto-MS/MS mode, using the following parameters: preferred charge states: 2–4, cycle time 3 s, MS spectra rate: 2 Hz, MS/MS spectra rate: 2 Hz to 6 Hz. MS/MS active exclusion was set after one spectrum unless the intensity increased fivefold. Collision energy was automatically calculated based on m/z and charge states.

Intact polar lipids (IPL) were analysed using HPLC on an HP 1100 liquid chromatography module (Agilent) coupled to an HCT mass spectrometer (Bruker) equipped with an electrospray ionisation (ESI) source used in positive and negative ionisation mode. The parameters of the ion source were as follows: source temperature 350 °C; capillary transfer voltage +5000 V (negative mode) and −4000 V (positive mode); nebulizer pressure 43 psi; drying gas flow rate (N₂) 8 L/min; drying temperature 350 °C; corona discharge 4 μA. The detection range (m/z ratio) was 500 to 3000 m/z. Nitrogen was generated from pressurised air by a Calypso 25.0 (F-DGSi) nitrogen generator and used as a cone gas and nebulisation gas. Chromatographic separation was carried out using an Inertsil Diol HILIC column (2.1 × 250 mm, 5 μm; GL Science) thermostated at 30 °C and equipped with a precolumn having the same stationary phase. The separation was carried out in solvent gradient mode and included as the mobile phase solvent A (isopropanol/water/formic acid/aqueous ammonia (88:10:0.12:0.04 v/v/v/v) and solvent B (n-heptane/isopropanol/formic acid/aqueous ammonia (79:20:0.12:0.04 v/v/v/v)). Lipids were eluted at a constant flow rate of 0.2 ml/min in a stepwise gradient from 100% solvent B to 66% solvent B over 18 min, then maintained for 12 min, followed by 35% solvent B in 15 min, maintained for 15 min and then back to 100% solvent B in 2 min, followed by equilibration for 20 min. The IPLs were solubilised in 1 mL of solvent B then 50 μl of an internal standard solution (1,2-Distearoyl-sn-glycero-3-phospho-rac-(1-glycerol) sodium salt; Sigma Aldrich; 100 μg/mL) were added. Injections of 10 μL were performed using an Agilent autosampler. Analyses were processed using Data Analysis software (version 4.2, Bruker Daltonics).

## Statistics and reproducibility

Nano-LC-MS/MS data were processed using FragPipe, a software based on MSFragger[82] for peptide identification and IonQuant[83] for label-free quantification. Spectral peptide matching was carried out with the following parameters: (1) mass tolerance of 10 ppm on the parent ion, (2) mass tolerance of 0.05 Da for fragmented ions from MS/MS, (3) carbamido-methylated Cys (+ 57.0215) as fixed modifications; and (4) oxidized Met (+ 15.9949), deamidated Asn and Gln (+ 0.9840) variable modifications. Proteins were then filtered with FDR < 1% (corresponding to a −10logP score above 25) for peptide-spectrum matches (PSMs) and a valid protein identification required minimum 2 unique and significant peptides.

For peptide quantification, the following discriminant parameters were used: (1) Intensity Type = MaxLFQ, (2) *p*-values cutoff = 0.05, (3) log2 fold change = 1, (4) Normalisation type = Variance stabilizing normalisation, (5) Imputation type = Perseus-type, (6) FDR correction = Benjamini Hochberg.

For all analyses, the presence of a given protein in a specific condition (brine type, UV-irradiation) was validated if it was properly identified in at least three biological replicates out of five. Post translational modifications were analysed using the PTM-Shepherd in FragPipe[84].

For LC-MS/MS data, protein identifications were performed using PEAKS® X-Pro software (64 bits version, 2020, BioInformatics solutions). It allows de novo assisted database search against the protein coding sequences from *H. salinarum* NRC-1. Proteins sequences were downloaded from UniProt (https://www.uniprot.org/proteomes/UP000000554).

For LC-MS/MS data, spectral peptides matching was carried out by following the same discriminant parameters as described for Nano-LC-MS/MS analyses. Proteins label-free quantification was performed using Peaks Q module with the following parameters: (1) Peptide count = 2, (2) Average area = 10E4, (3) Charges = 2-5, (4) Confident number samples per group = 2, (5) *p*-value = 0.01, (6) Fold change = 2, (7) Quality = 10, (8) FDR = 1%, (9) Significance method = ANOVA.

## Reporting summary

Further information on research design is available in the Nature Portfolio Reporting Summary linked to this article.

## Data availability

The authors declare that the data supporting the findings of this study are available within the paper and its Supplementary Data files. The data underlying Figs. 2, 3 and 5 can be found respectively in Supplementary Data 1, 2 and 3. Mass spectrometry proteomics data that were deposited to the ProteomeXchange Consortium via the PRIDE partner repository as datasets PXD062270 for data from *H. salinarum* cell envelope extracts incubated in the various brines with and without exposure to UV radiation, and PXD062272 for data from *H. salinarum* cultures acclimated to an Early Earth brine with and with exposure to UV radiation.

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

## Acknowledgements

This work was supported by financing to AK from the ANR project ExocubeHALO (ANR-21-CE49-0017) and the CNES Exobiology program (project Exocube). The authors gratefully acknowledge Giovanni Chiappetta from the Biological Mass Spectrometry and Proteomics Laboratory of ESPCI for Orbitrap nano-LS-MS/MS analyses. The MNHN mass spectrometers have been funded by the MNHN, the CNRS and the Région Ile-de-France. The authors also thank Caryn Evilia for the strain of *Halobacterium salinarum*. A.E. and R.B. gratefully acknowledge funding from the Deutsche Forschungsgemeinschaft (DFG, project ExocubeHALO, grant 490702919), from the Ministry of Economics and Energy, Germany (Projektraeger Deutsches Zentrum für Luft-und Raumfahrt, grants 50WB2023 and 50WB2323) and from Volkswagen Foundation and its Freigeist Program. Graphical abstract figure was partially created in BioRender[85].

## Author contributions

LB provided in project design, performed the experiments, analysed the data, and wrote the draft manuscript. AM and RP provided technical expertise, participated in mass spectrometry experiments and assisted in data analyses; SB provided technical expertise and helped perform NanoDSF experiments; PS performed intact polar lipid HPLC analyses; MT participated in acclimation experiments. RN provided key technical assistance for UV exposure experiments; AE participated in experimental design and data analyses, and project co-funding; AK provided funding and supervision for project design, execution, and manuscript preparation. All authors revised and approved the final manuscript.

## Competing interests

The authors declare no competing interest.
