## [Transparent Peer Review file · Communications Biology]

The salty tango of brine composition and UV photochemistry effects on *Halobacterium salinarum* cell envelope biosignature preservation

Corresponding Author: Dr Adrienne Kish

Version 0:

Reviewer comments:

Reviewer #1

(Remarks to the Author)

The authors of this manuscript are focused on determining the use of cell membrane lipids and proteins for their biosignature potential in fluid brine inclusions. In addition, archaea possess S-layers that have mineral-binding capabilities and thus, potential to serve as biosignatures. The goal of the reported work is to determine the impacts of brine composition and UV photochemistry has on the biosignature potential of the cellular membrane components of *Halobacterium salinarum* in brines mimicking brines on early Earth and Mars environments. Overall, the experimental plan covered the differences in ion composition based on likely environments that could yield evaporites, how possible chaotropic effects can impact protein stability, how protein compositions changed based on the brines they were exposed to, and the impact of UV radiation on proteins under the different brine compositions. In addition, the authors also attempted to explore how proteins from cells acclimated to early Guadaluoian Earth conditions responded to these conditions.

Overall, this manuscript addresses an important concern about the longevity of possible biosignatures in evaporite fluid brine inclusions under possible brine concentrations and exposure to UV radiation. Their experimental design was done well. Their conclusions were supported with the results they obtained. It is evident that much thought and planning went into this work as well as good analysis of the results. This will be helpful for researchers studying materials entrapped in evaporite fluid inclusions.

There are only a few corrections that need to be made.

1. In your Supplemental Figure #4, either provide the data for M3 or indicate that it appeared the same as the data for M4.
2. In line 373, Supplementary Table 6 is mentioned. There is no Supplementary Table 6.
3. In line 424, Supplementary Table 7 is mentioned but it does not appear in the Supplemental Material.
4. It would have been interesting if the authors could have mentioned in their discussion any known impacts of brines or UV radiation on S-layers of Archaea as it was mentioned in their Introduction but not discussed further.

Reviewer #3

(Remarks to the Author)

This manuscript provides a great example of the complexities in studying the interactions of multicomponent brines with biomolecules. This line of research follows a positive trend towards analysing the effects of environmentally relevant brine compositions instead of single component brines. The effects of UV radiation in combination with brines provides great insight into the complexities involved in detecting biosignatures

I mostly have minor comments, but have provided comments on the nanoDSF results which do not affect the interpretation of the major findings in this manuscript.

Comments:

Line 111 – What was the concentration of the Tris-HCl?

Line 128: For Type IV Early Mars, presumably this concentration of perchlorate is meant to be equivalent to a fluid inclusion within a crystal? As the bulk concentration of perchlorate in the Phoenix site is significantly lower.

Table 1: A couple of the oxygens appear to have been formatted as zeros (0 vs O)

Line 315: If brine 4 is indeed 4M Mg(ClO₄)₂ I'd be surprised if any protein could remain folded at such concentrations. Additionally any lipid association/ stability will be drastically affected.

Figure 2: For clarity, would 100% equate to detecting every known protein in the proteome from UniProt?

Line 429 – 431: With that logic, would one not expect the ratios of M4,E2, and E1 to be significantly below 1 if there was significant modification to the point of being undetectable? Does a ratio of 1 not mean that there was equally detectable PTMs between control and irradiated?

Line 471: The captions for (a) and (b) seem swapped, (a) is lipid in the figure but protein in the caption and vice versa.

Line 528-529: It's possibly less speculative to instead phrase this as the Trp residues were less hydrated. As membrane proteins don't necessarily have the same hydrophobic packing of cytosolic proteins, it could be that Trp is instead more solvated by lipids as opposed to a protein core/ interior, or by changes in the local association of neighbouring proteins in envelope fragments.

NanoDSF results: The application of nanoDSF to crude protein mixtures is an attractive approach that however comes with complications to the analysis which is made further complex in the study of envelope/ membrane associated proteins. The technique was designed largely for biopharmaceutical use as it requires significantly less material than a DSC measurement and is based on the central assumption that the change in fluorescence is caused by the increased exposure of hydrophobic residues to the solvent, followed by a decrease in fluorescence ascribed to protein aggregation at higher temperatures.

As membrane/ envelope proteins don't necessarily bury their hydrophobic residues, instead exposing them to the lipid acyl chains or at the headgroup/solvent interface, changes in fluorescence could be ascribed to changes in solvent/lipid/ intra or interprotein associations.

The clear peaks observed in BSS and BSS-LS are clear transitions, but likely resulting from the contributions of a small group of highly expressed proteins. Therefore the ascription of kosmo/chaotropicity may be limited to complex salt effects on a specific subset of proteins.

It has been shown previously that in crude extracts relatively small amounts of overexpression are needed to distinguish such proteins from the noise of the extract:

nanoDSF as screening tool for enzyme libraries and biotechnology development

<https://febs.onlinelibrary.wiley.com/doi/10.1111/febs.14696>

High-Throughput Feasible Screening Tool for Determining Enzyme Stabilities against Organic Solvents Directly from Crude Extracts

<https://chemistry-europe.onlinelibrary.wiley.com/doi/10.1002/cbic.201700526>

The same has been described for DSC analyses of crude cytosolic proteomes, where the major melting transitions observed are due to the melting of the ribosome, constituting ~30% of the soluble proteome.

In the analysis I would be hesitant to call the M1 brine kosmotropic, the steeper increase in fluorescence from 30-55C compared to BSS/ BSS-LS may be the result of a relatively uncooperative transition.

It's definitely interesting to see how nanoDSF reports on envelope protein stability, especially in the BSS and BSS-LS conditions, but the results also show the challenges that can come in interpreting DSF data of complex mixtures.

Version 1:

Reviewer comments:

Reviewer #1

(Remarks to the Author)

The authors of this manuscript focused on determining the use of cell membrane lipids and proteins for their biosignature potential in fluid brine inclusions. In addition, archaea possess S-layers that have mineral-binding capabilities and thus, potential to serve as biosignatures. The goal of the reported work is to determine the impacts of brine composition and UV photochemistry has on the biosignature potential of the cellular membrane components of *Halobacterium salinarum* in brines mimicking brines on early Earth and Mars environments. Overall, the experimental plan covered the differences in ion composition based on likely environments that could yield evaporites, how possible chaotropic effects can impact protein stability, how protein compositions changed based on the brines they were exposed to, and the impact of UV radiation on proteins under the different brine compositions. In addition, the authors also attempted to explore how proteins from cells acclimated to early Guadalupean Earth conditions responded to these conditions.

Overall, this manuscript addresses an important concern about the longevity of possible biosignatures in evaporite fluid brine inclusions under possible brine concentrations and exposure to UV radiation. Their experimental design was done well. Their conclusions were supported with the results they obtained. It is evident that much thought and planning went into this work as well as good analysis of the results. This will be helpful for researchers studying materials entrapped in evaporite fluid inclusions.

All of the suggested revisions were completed. I have no other concerns or comments.

Reviewer #3

(Remarks to the Author)

I thank the authors for considering the peer review comments. I am satisfied with the edits made.

	Comment	Revision
Reviewer 1		
1	In your Supplementary Figure #4, either provide the data for M3 or indicate that it appeared the same as the data for M4.	Due to the colour of the iron-rich M3 brine, the absorbance measurements were compromised with the microplate assay. Therefore, results could not be interpreted with cell envelope extract samples incubated in this brine. - Modified text, see Line 25-27: “The colour of the iron-rich M3 brine was found to be incompatible with the microplate assay due to both the high absorbance of this solution at 500 nm and the high chaotropicity of the brine which did not result in agar precipitation under the temperature range used.”
2	In line 373, Supplementary Table 6 is mentioned. There is no Supplementary Table 6.	Supplementary Table 5-7 were provided as an additional Excel files in the original submission
3	In line 424, Supplementary Table 7 is mentioned but it does not appear in the Supplemental Material.	Supplementary Table 5-7 were provided as an additional Excel files in the original submission
4	It would have been interesting if the authors could have mentioned in their discussion any known impacts of brines or UV radiation on S-layers of Archaea as it was mentioned in their Introduction but not discussed further.	Modified text, see Line 61-67: “The stability of S-layer interactions within the Halobacterium cell envelope, both between protein monomers and associations with lipids, is also dependent on ionic interactions ²⁷ . Both low salinity (NaCl) and chaotropic agents such as Li ⁺ and Mg ²⁺ can disrupt S-layer stability. In contrast, species adapted to more

		chaotropic magnesium-containing brines such as Haloferax volcanii require roughly 100 mM Mg²⁺ to stabilize their S-layer structure ²⁸. It is important to note that S-layer stability differs between whole cells and cell envelope extracts ²⁹, likely due to ion accessibility.” Modified text, see line 517-520: “As previously mentioned, incubation of H. salinarum cell envelopes in the acidic iron-rich M3 brine resulted in an absence of extractable membrane proteins. Instead, a brown precipitate formed during incubation, likely due to the high Fe²⁺ content of the brine. This can lead to ionic interactions with the negatively charged surfaces of S-layer proteins resulting in mineral formation in a process known to occur in the presence of excess sulphates, carbonates, or phosphates .”
Reviewer 3		
1	Line 111 – What was the concentration of the Tris-HCl?	Modified text, see Line 116 – “The resulting cell envelope pellets were then resuspended in 1 mL 0.1 M Tris-HCl buffer, pH 7.4 (Tris buffer)...”
2	Line 128: For Type IV Early Mars, presumably this concentration of perchlorate is meant to be equivalent to a fluid inclusion within a crystal? As the bulk concentration of perchlorate in the Phoenix site is significantly lower.	The concentration of perchlorate used for the M4 brine is indeed equivalent to a fluid inclusion within a crystal. - Modified text, see Line 133: Simulated Fluid Inclusion Brine Composition

3	Table 1: A couple of the oxygens appear to have been formatted as zeros (0 vs O)	Typographical errors have been fixed.
4	Line 315: If brine 4 is indeed 4M Mg(ClO ₄) ₂ I'd be surprised if any protein could remain folded at such concentrations. Additionally, any lipid association/ stability will be drastically affected.	The concentration of magnesium perchlorate in the M4 brine was indeed expected to lead to complete unfolding of the proteins as much lower perchlorate concentrations have been shown in previous studies to be detrimental to microorganisms' survivals. Additionally, this is what was observed with the NanoDSF data presented in our study. These data showed that no transition was observed in our study for cell envelope extracts incubated within M4 brine, presumably due to already unfolded proteins. Finally, upon incubation of the cell envelope extracts within the iron-free M4 brine, a decreased red colour intensity was observed almost instantly, indicating destabilization and/or degradation of the carotenoid lipid pigments that produce this characteristic colouration of haloarchaea. - Modified text, see Line 319-325: "The M4 brine (Fig. 1e) did not show any observable transition. It is hypothesized that the chaotropic nature of brines E2, M2, and M4, as assessed by the agar gel-point assay, induced partial or complete denaturation of cell envelope proteins during the incubation process prior to

		measurement. This likely resulted in minimal or even a lack of transitions between folded and unfolded states during the DSF assay. This phenomenon was anticipated for the M4 brine, given the well-characterized strong chaotropic properties of perchlorate. Upon incubation of cell envelope extracts in the M4 brine, a rapid decrease in the intensity of their typical red colour was observed, indicating the destabilization or degradation of carotenoid lipid pigments responsible for the characteristic red-orange colouration of haloarchaea.”
5	Figure 2: For clarity, would 100% equate to detecting every known protein in the proteome from UniProt?	The text was modified to clarify this point as follows: - Modified text, see Line 363-367: “Assessment of the effect of brines on cell envelope membrane protein composition through nano-LC/MS-MS. The mismatch percentages represent the number of proteins that were uniquely or not identified from cell envelope extracts incubated in the respective brines compared to cell envelope incubated in the control brine, BSS. Refer to section 2.6.3 for the discriminant parameters for protein identification. (a) Cell envelope proteins; (b) Cell envelope proteins + cytosolic contaminants.”

6

Line 429 – 431: With that logic, would one not expect the ratios of M4,E2, and E1 to be significantly below 1 if there was significant modification to the point of being undetectable? Does a ratio of 1 not mean that there was equally detectable PTMs between control and irradiated?

A ratio of 1 indicates equal levels of detectable UV-induced damages between the control and irradiated samples. It was expected that irradiation would lead to increased protein modifications due to interactions with UV light and reactive species or radicals, resulting in a ratio greater than 1. However, the ratios observed for the brines that caused the highest degradation of membrane proteins under UV irradiation (M4, E1, and E2) were not significantly higher as anticipated. This was hypothesized to result from the complete degradation of modifications-bearing peptides, rendering the modifications unidentifiable by mass spectrometry analysis. Consequently, since the modifications caused by irradiation were not fully captured, the number of detectable modifications in the irradiated samples was similar to that in the non-irradiated samples, leading to a ratio close to 1.

-

Modified text, see Line 437-442: “In contrast, for brines that induced higher protein degradation under UV irradiation (E1, E2, M1, and M4; Fig. 5b), most, especially M4 and E2, did not generate abundant photo-induced damages to cell envelope proteins upon UV radiation, with

		their respective ratios also close to 1. Although those brines showed high production of ROS, this phenomenon may be explained by potential extensive degradation of modified peptides, rendering them undetectable by mass spectrometry. Therefore, the number of photo-chemical modifications detected in the irradiated and control samples were similar, leading to ratio close to 1.
7	Line 471: The captions for (a) and (b) seem swapped, (a) is lipid in the figure but protein in the caption and vice versa.	Error has been fixed. - Modified text, see Line 483-487: (a) Main differences in lipid composition between acclimated and control cell envelopes. Lipids were identified by LC-MS and analysed using DataAnalysis (Bruker). Semi quantification was performed using (1,2-Distearoyl-sn-glycero-3-phospho-rac-(1-glycerol) as internal standard. (b) Comparison of cell envelope protein composition between acclimated and control cell envelopes. Proteins were analysed by mass spectrometry and treated using PeaksX Studio
8	Line 528-529: It's possibly less speculative to instead phrase this as the Trp residues were less hydrated. As membrane proteins don't necessarily have the same hydrophobic packing of cytosolic proteins, it could be that Trp is instead more solvated by lipids as opposed to a protein core/ interior, or by changes in the local association of neighbouring proteins in envelope fragments.	Modified text, see Line 544-547: "The Trp emission spectra of cell envelopes incubated in the M2 brine corresponded to a greater number of Trp residues solvated within the lipid membrane structure compared to incubation in the control BSS

		brine. In contrast, Trp residues of cell envelope proteins incubated in the M4 brine became more exposed to the brine environment.”
9	NanoDSF results: The application of nanoDSF to crude protein mixtures is an attractive approach that however comes with complications to the analysis which is made further complex in the study of envelope/membrane associated proteins. The technique was designed largely for biopharmaceutical use as it requires significantly less material than a DSC measurement and is based on the central assumption that the change in fluorescence is caused by the increased exposure of hydrophobic residues to the solvent, followed by a decrease in fluorescence ascribed to protein aggregation at higher temperatures. As membrane/ envelope proteins don't necessarily bury their hydrophobic residues, instead exposing them to the lipid acyl chains or at the headgroup/solvent interface, changes in fluorescence could be ascribed to changes in solvent/lipid/ intra or interprotein associations. The clear peaks observed in BSS and BSS-LS are clear transitions, but likely resulting from the contributions of a small group of highly expressed proteins. Therefore the ascription of kosmo/chaotropicity may be limited to complex salt effects on a specific subset of proteins. It has been shown previously that in crude extracts relatively small amounts of overexpression are needed to distinguish such proteins from the noise of the extract: nanoDSF as screening tool for enzyme libraries and biotechnology development https://febs.onlinelibrary.wiley.com/doi/10.1111/febs.14696 High-Throughput Feasible Screening Tool for Determining Enzyme Stabilities against Organic Solvents Directly from Crude Extracts    As mentioned, the samples and salt conditions used in this study are far more complex than those traditionally analysed with NanoDSF. Applying this technique in such a setting presented significant challenges, uncovering complexities that required correlation with other techniques for proper interpretation. The reviewer's concern regarding the influence of a small group of highly expressed proteins is highly relevant. In these samples, the highly expressed proteins included transmembrane proteins such as transducers and cell surface glycoproteins (S-layer). Therefore, the observed transitions partially reflect the impact of the different brines on the cell envelope. Future studies should explore additional methods to complement these findings, providing a clearer understanding of the chaotropic and kosmotropic effects on the cell envelope. Testing alternative methods would also help clarify the specific effects of the M1 brine. - Modified text, see Line 640-647: “Moreover, this study also highlights the complexity of applying NanoDSF to highly 	

europe.onlinelibrary.wiley.com/doi/10.1002/cbic.201700526

The same has been described for DSC analyses of crude cytosolic proteomes, where the major melting transitions observed are due to the melting of the ribosome, constituting ~30% of the soluble proteome. In the analysis I would be hesitant to call the M1 brine kosmotropic, the steeper increase in fluorescence from 30-55C compared to BSS/ BSS-LS may be the result of a relatively uncooperative transition. It's definitely interesting to see how nanoDSF reports on envelope protein stability, especially in the BSS and BSS-LS conditions, but the results also show the challenges that can come in interpreting DSF data of complex mixtures.

intricate samples and salt conditions, far exceeding the simplicity of traditional setups. While the technique offered valuable insights, its limitations necessitated complementary analyses to interpret the observed phenomena. Furthermore, the observed transitions could be influenced by a small number of highly expressed proteins affecting the overall fluorescence response, as previously demonstrated⁸¹. Given that transmembrane transducer proteins and S-layer glycoproteins were highly abundant in the cell envelope extract, NanoDSF provides valuable insights. However, the information obtained may be incomplete. Future research should prioritize integrating alternative methods to expand upon these findings, providing a more comprehensive picture of the chaotropic and kosmotropic effects on cell envelope stability, particularly in challenging brine conditions.”